# Environmental Exposures Increase Health Risks in Childhood Cancer Survivors

**DOI:** 10.3390/cancers17132223

**Published:** 2025-07-02

**Authors:** Omar Shakeel, Nicole M. Wood, Hannah M. Thompson, Michael E. Scheurer, Mark D. Miller

**Affiliations:** 1Texas Children’s Cancer and Hematology Center, Baylor College of Medicine, Houston, TX 77030, USA; 2Children’s Mercy Kansas City, University of Missouri-Kansas City School of Medicine, Kansas City, MO 64108, USA; nwood@cmh.edu; 3Department of Environmental Medicine and Climate Science, Icahn School of Medicine at Mount Sinai, New York, NY 10029, USA; hannah.thompson@mssm.edu; 4Children’s Healthcare of Atlanta, Emory University School of Medicine, Atlanta, GA 30322, USA; michael.scheurer@emory.edu; 5Western States Pediatric Environmental Health Specialty Unit, University of California, San Francisco, CA 94143, USA

**Keywords:** environmental health, pediatric cancer, environmental referral service, childhood cancer survivors

## Abstract

This communication reviews how environmental exposures to air pollution, tobacco smoke, extreme weather events, and pesticides can negatively affect the health and survival of childhood cancer survivors. We recommend the development of an environmental referral service to help address and decrease these risks. Including environmental health in pediatric cancer care will help support healthier choices, inform families, and improve long-term outcomes for childhood cancer patients.

## 1. Introduction

Childhood cancer survivors (CCSs) are a vulnerable population at risk of chronic health conditions due to late effects of cancer and its treatment. The growing number of survivors of childhood cancers has led to an increase and acceleration in life-long and chronic health issues. By age 50, 99% of childhood cancer survivors have at least one chronic health problem [1]. Global studies have shown that survival and health outcomes among children, adolescents, and young adults with cancer are impacted by social determinants of health, including the environment to which they are exposed [2,3,4,5].

There has been growing concern over the impact of environmental exposures, such as air pollution, tobacco smoke, extreme weather, and pesticides on the outcomes and general health of pediatric cancer survivors. Although these are not the only environmental exposures that can affect the morbidity and mortality of CCS, they are among those most directly supported by evidence in the existing literature. However, there is a knowledge gap present among providers in discussing the impact of environmental exposures on the health of pediatric cancer patients. We review the existing literature on these specific exposures and recommend the development of an environmental referral service to address these concerns by providing evidence-based mitigation strategies, potentially contributing to improved health outcomes and surivival in childhood cancer survivors. This communication presents emerging evidence and a practical proposal for addressing environmental risk in childhood cancer survivorship.

## 2. Environmental Exposures Are Linked to Health Outcomes in Childhood Cancer Survivors

Air pollution is a significant environmental exposure that adversely affects the health of cancer survivors. Living in areas of high air pollution, defined as exposure to fine particles ≤ 2.5 microns in diameter, PM_2.5_, has been significantly associated with decreased survival among children with cancer, regardless of socioeconomic status [6]. Another study of pediatric cancer survivors in Utah reported that a higher 3-day average PM_2.5_ by zip code was associated with an increased odds of respiratory hospitalization (OR: 1.84; 95% CI: 1.13–3.00) [7]. In a follow-up study, the authors also reported that cumulative zip code-level estimates of PM_2.5_ in ambient air were positively associated with all-cause pediatric cancer mortality among survivors who had lymphoid leukemia, lymphoma, or brain tumors at five and ten years post diagnosis [8]. There were also significant associations between PM_2.5_ and mortality among adolescent and young adult cancer survivors [8]. Air quality standards in the United States are determined by the Environmental Protection Agency (EPA) and were recently revised in February 2024, lowering the air quality standard for fine particulate matter (PM_2.5_) from 12.0 to 9.0 µg/m^3^ [9]. Despite this, residing in zip codes with PM_2.5_ levels greater than or equal to the new EPA air quality standard was also significantly associated with worse overall survival among children with cancer [6]. Another recent study found that childhood survivors of acute myeloid leukemia and hepatoblastoma in Texas experienced lower survival rates when residing near an oil or gas well at the time of diagnosis, likely due to exposure to hazardous air pollutants emitted from these sites [10]. Despite caregivers’ interest in learning more about the effects of air pollution on survivors’ health, there remains limited awareness, unsuccessful information-seeking, and minimal exposure reduction behaviors among caregivers of childhood cancer survivors [11].

Tobacco is a well-known environmental carcinogen and the leading cause of preventable death worldwide. Unfortunately, one in four childhood cancer survivors smokes [12]. Given that childhood cancer survivors are likely to have existing cardiovascular and respiratory health issues, they are at even greater risk of treatment-related morbidities and early mortality from tobacco use [13]. In one study, tobacco use increased lung cancer risk more than 20-fold in Hodgkin lymphoma survivors [14]. Another study found that paternal smoking prior to conception was associated with reduced survival in children with acute myeloid leukemia [15]. Similarly, exposure to secondhand smoke has been associated with decreased survival and increased treatment-related mortality in children with acute lymphoblastic leukemia [16].

Extreme weather events pose an emerging global threat to pediatric cancer survivors by increasing exposure to environmental carcinogens and affecting cancer survival through impediments to access and delivery of care [17,18,19,20]. Extreme weather can discrupttransportation, communication, and power systems leading to delays in patient care. Sociodemographic factors, along with chronic physical and mental health issues, may predispose pediatric cancer survivors to increased vulnerability to shifting weather patterns and extreme weather events [18].

Pesticides are another important environmental toxicant used worldwide. Systematic reviews of studies have shown that pesticides are associated with poor cognitive, behavior, and motor neurological outcomes in children [21,22]. Childhood cancer survivors are especiallysusceptible to cognitive and behavioral issues because of their cancer and associated treatment;exposure to pesticides may exacerbate these risks. Although several studies have shown that pesticide exposure in children is associated with an increased risk of hematopoietic and central nervous cancers, evidence remains limited regarding the impact of such exposures on the health outcomes of childhood cancer survivors [23,24,25]. A 2019 study showed that agricultural occupational exposure to pesticides was associated with treatment failure in adults with diffuse large B-cell lymphoma [26]. A 2025 study was the first to identify reduced survival among children with acute lymphoblastic leukemia who were exposed to residential pesticides, particularly rodenticide, during pregnancy [25].

## 3. Addressing Environmental Health in Childhood Cancer Survivors

### 3.1. Survey and Focus Group

In April 2024, an institutional REDCap survey was sent via email to approximately 154 members of the oncology team at Texas Children’s Cancer and Hematology Center, a major pediatric cancer hospital in the United States. The survey was not validated prior to distribution, and IRB and ethical approval was not necessary. The survey included questions on knowledge of environmental risk factors for pediatric cancer and interest in using an environmental referral service if one were developed. The data were exported from REDCap and analyzed using qualitative methods.

A one-hour focus group discussion involving nineteen members from the Cancer and Hematology Center Family Advisory Team was then conducted and transcribed.. The Family Advisory Team consisted of family members who voluntarily indicated their interest through an engagement intake form and had a child currently undergoing or who had completed cancer treatment. The discussion revealed a strong need for the development of a pediatric cancer environmental referral service. Many families expressed frustration that healthcare providers are not prepared to address these environmental issues. Additionally, they said that they are looking for this information and currently left to search the internet, which can lead to misinformation and may be detrimental during and after their child’s cancer treatment. Overall, families felt that integrating such a service into standard clinical practice would significantly reduce the anxiety that many of them experienced during their cancer journeys, including survivorship.

### 3.2. Survey Results

There were 49 respondents to the survey after a two-week period. In total, 30% of respondents were physicians, 15% were advanced practice providers, 32% were nurses, and 23% were social workers. Approximately 80% of respondents indicated that they receive questions from families on the impact of the environment on their child’s cancer. While most reported receiving questions, nearly 75% were not comfortable discussing the topic with families. Ninety-six percent of respondents indicated that an environmental referral service would be helpful, with most expressing interest in using the service to address concerns in addition to or in lieu of discussing them directly with families. Approximately 90% of respondents indicated that they would not have any concerns using an environmental referral service. Several respondents indicated that a referral service would be helpful to incorporate into clinical practice. Given that data are self-reported and derived from a single-center sample, results should be interpreted with caution due to potential response bias and lack of generalizability (Figure 1).

## 4. Discussion

Given the significant impact of environmental exposures on pediatric cancer survivors, it is important to address the limited knowledge, familiarity, and comfort of healthcare providers in discussing these topics with patients and families [27]. In response to the survey results and focus group discussion, we recommend a practical and scalable intervention to integrate environmental health into pediatric cancer care.

United States pediatric cancer environmental referral services are now under development by the Childhood Cancer and the Environment Program of the Pediatric Environmental Health Specialty Units (PEHSUs), a national network of experts in pediatric health issues that arise from environmental exposures (https://pehsu.net/program/childhood-cancer-the-environment-national-program/, accessed on 5 June 2025). The referral service will work with the PEHSU to help address the questions and concerns from patients and families and provide up-to-date and evidence-based information about environmental risk factors in addition to simple and cost-effective strategies to reduce exposure risk. Institutions may approach the integration of environmental health in pediatric cancer care differently. The referral service will be adaptable to meet this need and may be initiated at various points along the care continuum—including cancer predisposition and screening clinics, initial diagnosis, and throughout survivorship care. In Europe, the PEHSU in Murcia, Spain, has already launched the Environmental and Community Health Program for Longitudinal Follow-up of Childhood and Adolescent Cancer Survivors [28]. This program includes a pediatric environmental history tool to assess individual risk and develop low-carbon, healthy lifestyle strategies [28,29,30]. This model was presented at the Childhood Cancer Prevention Symposium held in Houston, Texas, in February 2025, where it was received with significant interest as a scalable approach to integrating environmental health into pediatric oncology care. The model may be adaptable across diverse healthcare settings nationally and internationally, building on programs like the PEHSU in Murcia, Spain.

An environmental health history and risk assessment form is also under development by the Childhood Cancer and the Environment Program and will be utilized in conjunction with the referral service. The intake form will capture the geographical locations of the patient up until their diagnosis along with details about siblings and birth order, diet and nutrition, and exposures to tobacco, indoor and outdoor air pollution, chemicals and pesticides, and UV/radiation before and after cancer diagnosis. Responses will be confidential and incorporated into the secure electronic health record as part of the patient’s social and environmental history. Based on the responses to the intake form, the referral service will offer tailored resources and counseling on protective measures to reduce risk within the home and community. As part of these resources, an environmental toolkit will be offered to families and will include affordable air quality and carbon dioxide monitors, as well as a DIY air purifier made from a box fan and HEPA filter.

## 5. Conclusions

We recommend the development of a dedicated pediatric cancer environmental referral service within pediatric oncology cancer centers to address environmental exposures as important and potentially modifiable risk factors in pediatric cancer patients. Healthcare providers and families have expressed a clear need for a reliable resource to help navigate environmental health concerns and mitigate risks. Integrating environmental health into pediatric cancer care would provide a valuable opportunity to address environmental exposures, empower patients and families by promoting healthier behaviors, and potentially improve long-term health outcomes by reducing morbidity and mortality in this vulnerable population.

## Figures and Tables

**Figure 1 cancers-17-02223-f001:**
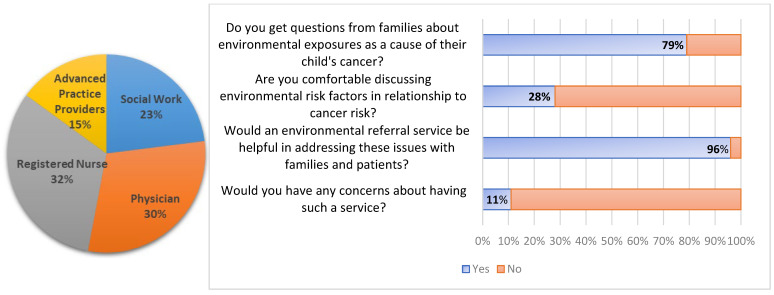
Distribution of oncology team members who responded to the survey (N = 49) and their experiences with receiving questions from families on environmental risk factors for pediatric cancers and interest in using an environmental referral service.

## Data Availability

The original contributions presented in this communication are included in the article. Further inquiries can be directed to the corresponding author.

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
