# Peer review of "Environmental Exposures Increase Health Risks in Childhood Cancer Survivors"

_cancers, 2025, doi:10.3390/cancers17132223_

Round 1
Reviewer 1 Report
Comments and Suggestions for Authors
This communication reviews how environmental exposures to air pollution, tobacco smoke, extreme weather, and pesticides can negatively affect the health and survival of childhood cancer survivors.
The authors address the impact of environmental exposures, such as air pollution, tobacco smoke, extreme weather events, and pesticides, on the health outcomes of CCS. The bibliography is adequate.
In April 2024, an institutional REDCap survey was sent via email to members of the healthcare team at Texas Children’s Cancer and Hematology Center, a major pediatric cancer hospital in the United States. Approximately 80% of respondents indicated that they receive questions from families on the impact of the environment on their child’s cancer. While most reported receiving questions, nearly 75% were uncomfortable discussing the topic with families.
This article, therefore, includes the main publications in the literature on environmental toxicants. A study done in Texas found that families consider it a very important subject. However, health professionals are not sufficiently informed. A single service, the PEHSU in Murcia, Spain, has already launched the Environmental and Community Health Program for Longitudinal Follow-up of Childhood and Adolescent Cancer Survivors.
This article should be sent to former patient associations to insist on reducing exposure to these environmental toxicants.
Author Response
Thank you very much for taking the time to review this manuscript. We agree with your comments.
Reviewer 2 Report
Comments and Suggestions for Authors
This manuscript offers a significant and punctual examination of how environmental factors—specifically air pollution, tobacco exposure, pesticide use, and extreme climate events—affect childhood cancer survivors (CCS). The Authors introduce a novel proposal to set up an environmental referral service within pediatric oncology settings to address and reduce these risks. The topic is highly relevant, especially given the growing survivorship population and the increasing recognition of social and environmental determinants of health. The article is well structured, the proposal is innovative, and the supporting data from institutional surveys and family advisory groups are compelling.
However, the manuscript would benefit from minor to moderate revisions in order to improve clarity, structure, and scientific accuracy.
- Introduction
- The introduction offers an adequate overview of the topic but would benefit from more concise phrasing in specific sections.
- The Authors are encouraged to briefly highlight the current gap in clinical practice concerning environmental health discussions with CCS and their families.
- Methods
The survey administered to healthcare professionals (REDCap, April 2024) should be described more thoroughly in the Methods section. Specifically, please clarify:
- The selection criteria for participants;
- Whether the survey was validated or piloted before distribution;
- Any ethical considerations or Institutional Review Board approval obtained.
Similarly, the focus group conducted with members of the Family Advisory Council should be more thoroughly detailed:
- define how participants were selected;
- Indicate whether the sessions were recorded and/or transcribed;
- Explain the analytical approach used to interpret qualitative data.
- Structure and Content Adjustments
- The subsections “Data collection and processing” and the narrative on the “Focus Group” would be more appropriately placed in a dedicated Methods or Results section rather than inserted within the main body of the text.
- It would be helpful to explain the rationale for focusing on the selected environmental exposures (e.g., PM2.5, rodenticides, secondhand smoke). If a systematic review was conducted to support their inclusion, the methods used should be briefly described.
- Discussion
The discussion is insightful but remains too concise in places. It would benefit from:
- More in-depth comparisons between the study’s findings and existing literature or models used in other healthcare settings;
- Reflections on limitations, such as single-center survey data, potential response bias, and generalizability to different populations.
- Language and Style
- A thorough English language revision is advisable throughout the manuscript.
- Avoid redundant expressions and repeated use of terms within the same paragraph (e.g., "air pollution" repeated excessively).
- Improve transitions between paragraphs to enhance the narrative flow.
- Figures and Tables
- Figure 1 is informative but should include a more descriptive title and caption.
- Consider providing a summary table with key survey findings and response percentages for clarity.
- Ethical Considerations
- The manuscript should state whether ethical approval was obtained for the institutional survey and the family advisory focus group.
- Please clarify whether informed consent was required and collected.
Conclusion
This is a convincing and much-needed addiction to the field of pediatric oncology survivorship. With some improvements in methodological clarity, structure, and language, this manuscript could serve as an excellent model for integrating environmental health into long-term cancer care for children.
A comprehensive review of the English language throughout the manuscript is recommended.
-
-
Refrain from repeating the same terms within a single paragraph (e.g., repeated mentions of "air pollution").
-
Use clearer transitions to improve logical flow between sections and ideas.
-
Author Response
Please see attachment.
Thank you,
Omar

Reviewer 3 Report
Comments and Suggestions for Authors
American authors dedicated their study to the childhood cancer survivors (CCS) which are at increased risk for chronic health issues due to late effects of cancer and its treatment. Authors theoretically addressed the impact of environmental exposures, such as air pollution, tobacco smoke, extreme weather events, and pesticides on the health outcomes of CCS. Later, authors present a survey where they show that providers at a major pediatric cancer center in the United States have limited knowledge about environmental risk factors.
The article is interesting and topic is important. However, the structure of this work is unusual. It is neither a review , meta-analysis, nor prospective or retrospective study. This is mix of a short review, presentation of own survey and, finally, something like a presentation of next steps.
This last part is something I am not sure how it is or is not appropriate for such an article. When authors would present something they already sone (like survey) and discuss results, or when they would write a review only, with factors, impact and recommendation for further research and measures , this would be optimal.
Could authors rewrite the last part, making it as recommendation rather than a plan of next steps?
Author Response
Please see the attachment.
Thank you,
Omar

Round 2
Reviewer 3 Report
Comments and Suggestions for Authors
-